# Exploratory study of the global intent to accept COVID-19 vaccinations

Alexandre de Figueiredo [1✉] & Heidi J. Larson [1,2,3]

## Abstract

**Background** As the world begins the rollout of multiple COVID-19 vaccines, pandemic exit strategies hinge on widespread acceptance of these vaccines. In this study, we perform a large-scale global exploratory study to examine the levels of COVID-19 vaccine acceptance and explore sociodemographic determinants of acceptance.

**Methods** Between October 31, 2020 and December 15, 2020, 26,759 individuals were surveyed across 32 countries via nationally representative survey designs. Bayesian methods are used to estimate COVID-19 vaccination acceptance and explore the sociodemographic determinants of uptake, as well as the link between self-reported health and faith in the government's handling of the pandemic and acceptance.

**Results** Here we show that intent to accept a COVID-19 vaccine is low in Lebanon, France, Croatia, and Serbia and there is population-level polarisation in acceptance in Poland and Pakistan. Averaged across all countries, being male, over 65, having a high level of education, and believing that the government is handling the pandemic well are associated with increased stated acceptance, but there are country-specific deviations. A belief that the government is handling the pandemic well in Brazil and the United States is associated with lower vaccination intent. In the United Kingdom, we find that approval of the first COVID-19 vaccine in December 2020 did not appear to have an impact on the UK's vaccine acceptance, though as rollout has continued into 2021, the UK's uptake exceeds stated intent in large-scale surveys conducted before rollout.

**Conclusions** Identifying factors that may modulate uptake of novel COVID-19 vaccines can inform effective immunisation programmes and policies. Differential stated intent to accept vaccines between socio-demographic groups may yield insights into the specific causes of low confidence and may suggest and inform targeted communication policies to boost confidence.

## Plain language summary

The aim of this study was to understand what percentage of people would accept a COVID-19 vaccine in various countries across the world, and to understand what groups of people would be more or less likely to accept the vaccine. We analysed the response to surveys about people's intention to get a COVID-19 vaccine performed in 32 countries, and find that intent to accept a COVID-19 vaccine is comparatively low in Lebanon, France, Croatia, and Serbia. We also find that across all countries considered, being male, older, or having a high level of education is associated with increased likelihood to state a willingness to accept a COVID-19 vaccine. By understanding why different groups have differing opinions about the COVID-19 vaccine, we may be able to better understand specific concerns and assist healthcare policymakers to design more effective risk-communication strategies.

---

[1] Department of Infectious Disease Epidemiology, London School of Hygiene and Tropical Medicine, London, UK. [2] Department of Health Metrics Sciences, University of Washington, Seattle, WA, USA. [3] Centre for the Evaluation of Vaccination, Vaccine & Infectious Disease Institute, University of Antwerp, Antwerp, Belgium. ✉email: alex.defigueiredo@lshtm.ac.uk

The rollout of vaccines against the novel coronavirus disease (COVID-19) has begun to populations around the world. While large-scale vaccination manufacture, storage, supply, and delivery all present noteworthy logistical challenges for successful vaccination campaigns, addressing acceptance barriers to COVID-19 vaccines needs equal attention.

Prepandemic evidence from the Vaccine Confidence Project's (www.vaccineconfidence.org) multinational studies of confidence in vaccines suggests that older age groups are generally among the most confident in vaccines[1–4]. The initial rollout of vaccines to (predominately older age) high-risk groups across many settings is unlikely to meet substantial resistance. Emerging evidence from Public Health England and the Israeli Ministry of Health suggest suggest higher rate of breakthrough infections than was expected via clinical vaccine trials[5,6]; therefore, herd immunity may be "vaccine-assisted" in the sense that vaccines substantially reduce morbidity and mortality while population-level herd immunity builds up through natural infection. However, pandemic exit strategies are widely viewed as relying on achieving vaccination levels that exceed those required for herd/community immunity, which will require uptake from younger age groups, who are among the least likely to state willingness to accept a COVID-19 vaccine in a variety of settings, including France[7], Germany[8], Sweden[8], United States[9,10], and the United Kingdom[11,12] (though there is also evidence that *younger* age groups are more likely to accept the vaccine in Mexico[8]). In addition, other factors such as gender, education level, income, and ethnicity have been found to be associated with intent to vaccinate in a range of large studies[13]. To date, two multicountry studies exploring national intent to accept a COVID-19 vaccine appear in the published literature, exploring national-level differences in acceptance in June 2020 in 19 countries[14] and from March to September 2020 in 12 countries[8].

This study builds on the growing literature that explores potential uptake of a novel COVID-19 vaccine by widening the number of countries surveyed and by providing a more recent appraisal of vaccine acceptance in many settings. This study also includes countries that were surveyed after the Pfizer–BioNTech vaccine had been submitted for emergency use authorisation[15] and, in the United Kingdom, after this vaccine had both been approved for use[16] and the first patient vaccinated[17].

In this study, intent to accept a COVID-19 vaccine is explored for 26,759 individuals across 32 countries between October 21 and December 15, 2020, using data from the Worldwide Independent Network of Market Research (WIN) World Survey. As of 25 January 2021, these countries represent 73% of the total global mortality burden and among countries with recent and historic vaccine confidence issues such as France[18], Nigeria[19], Pakistan[20], Poland[21], United States[22], and the United Kingdom[23]. A range of possible drivers of COVID-19 vaccine acceptance are considered, including sociodemographic status (sex, age, highest education achieved, employment status, and income); self-reported health (overall health and stress); and perceptions around government handling of the pandemic. Country-level variables are used to explore trends at the national level. A comparison is made between the WIN World Survey data and 10,822 individuals surveyed in June 2020 in 15 of the same countries as surveyed here[14]. Time-varying trends in intent to accept a COVID-19 vaccine are assessed in the United Kingdom where previous survey data[11,24] allow a temporal comparison before and after the approval and introduction of the Pfizer–BioNTech vaccine in the United Kingdom.

We show that intent to accept a COVID-19 vaccine is relatively low in Croatia, France, Lebanon, and Serbia, and highest in Vietnam, India, China, and Denmark. We also find that being male, older, and having a high level of education is associated with higher intent to accept a COVID-19 vaccine, but there are country-level variations around these global trends. Individuals who believe that the government is handling the pandemic well tend to be more inclined to vaccinate, except in Brazil and the United States, where a belief that the government is handling the pandemic well is associated with lower intent to vaccinate.

## Methods

**Data**. Between October 31, 2020, and December 15, 2020, a total of 26,759 individuals aged 18 years or older across 32 countries were surveyed as part of WIN World Survey: Argentina, Brazil, Canada, Chile, China, Croatia, Denmark, Ecuador, Finland, France, Germany, Hong Kong, India, Indonesia, Ireland, Italy, Japan, Lebanon, Malaysia, Mexico, Nigeria, Pakistan, Paraguay, Peru, Poland, Serbia, Slovenia, South Korea, Spain, United Kingdom, United States, and Vietnam. The WIN World Survey is an international survey carried out by WIN every year to measure people's thoughts, expectations, and perceptions toward relevant topics for society[25]. Surveys are collected via online surveys, computer-assisted telephone surveys, and face-to-face interviews, depending on, for example, countries' internet penetration and COVID-19 limitations see supplementary Table 1 for further details on survey methodologies, sample sizes, and fieldwork dates. In the survey, respondents are asked "When a vaccine for the coronavirus becomes available, will you get vaccinated?" There are four possible responses (henceforth denoted $Y$ and modelled as an ordinal four-vector): "definitely will get vaccinated" (4), "unsure but probably will get vaccinated" (3), "unsure but probably will not get vaccinated" (2), and "definitely will not get vaccinated" (1). In addition, individual-level socio-demographic data on respondents' sex, age, highest education level, work status, and income quintile are collected as well as individuals' self-reported health ("how do you consider your overall health in general?") and stress ("how often would you say that you suffer from stress?"). Individuals' perceptions on how their government has handled the pandemic ("please rate […] the way your government [has] handled the [coronavirus] crisis") are also measured. To allow the statistical investigation into factors that may affect country-wide variation in vaccination intent, country-level data were collected on: the cumulative SARS-CoV-2 deaths per 100,000 population since the start of the pandemic until the beginning of survey fieldwork in each country (deaths, rather than cases, are used as deaths are likely a more robust measure of the state of an epidemic as testing capacities may vary more substantially across countries) and within the most recent two weeks before fieldwork[26]; the Human Development Index (HDI)[27]; GDP per capita (GDP)[28]; and confidence in the importance of vaccines for children[4]. Estimates of country-level confidence in the importance of vaccines for children[4] are used as a proxy for a country's overall vaccine confidence as they show a strong univariate association with parental acceptance of vaccines for their children (and because there is a high collinearity between national-level confidence in the importance of vaccines for children and in other national-level confidence measures for the safety and effectiveness of vaccines: collinearity may lead to inflated confidence intervals around parameter estimates[29]). The survey questionnaire is provided in the supplementary information. We note that some variables collected were not of interest to us in this exploratory study, for example, public views on the health-system capacities and self-reported rates of smoking or drinking.

During data collection, quotas aligning with national demographic distributions for age, sex, and subnational region were set with survey weights calculated when quotas are not met so that proportions of these demographic traits match national-level

distributions. All surveys—with the exception of Ecuador and Vietnam—are therefore nationally representative according to national-level sex, age, and region demographic distributions. In Ecuador and Vietnam, surveys took place in Quito-Guayaquil (Ecuador) and Ha Noi and Ho Chi Minh City (Vietnam) and quotas (and associated weights) were set to align with these subnational regions. Data are collected using online surveys (25 countries, $n = 20855$ respondents), computer assisted telephone interviews (four, $n = 2803$), telephone-assisted web interview (one, $n = 600$), and face-to-face interviews (two, $n = 2500$), see supplementary Table 1. Informed consent was obtained by WIN for all survey participants.

**Statistical analysis**. National-level intent to accept COVID-19 vaccines is estimated using a categorical distribution with a Dirichlet prior. The associations between individuals' uptake intent and individual- and country-level determinants are obtained via Bayesian multilevel regressions[30]. Intent to accept COVID-19 vaccines is related to sociodemographic data (sex, age, highest education level, work status, and income quintile) and self-reported health and stress, as well as perceptions on how their government has handled the pandemic and the country-level factors described in Data. Intent to accept a COVID-19 vaccine is related to these individual- and national-level covariate data via Bayesian ordinal multilevel logistic regressions (detailed below). A total of 95 respondents responded that they "do not know" or provided no response as to whether they would get a COVID-19 vaccine. To avoid the loss of missing data, these 95 responses are recoded to "unsure but probably will not get vaccinated" as they demonstrate some hesitancy about vaccinating, but no strong intent to reject the vaccine. (A sensitivity analysis is performed to test the robustness of this classification against other classification methods, and this does not impact our findings: see supplementary methods for further information).

There were five missing values for vaccination intent, these responses (all from Vietnam) were removed from the analysis as were seventeen unspecified values for covariate data (10 in Germany for income and seven for age in Pakistan). These missing data represent 0.09% of all responses. For all individual-level covariates, a "do not know/ no response" category was created for all covariates to again avoid the loss of missing data. The reference group for individual-level covariates is an employed, female, aged 18–24, with secondary education, in the middle-income group, and who self-reports as healthy, not often stressed, and thinks their government is handling the pandemic badly (see Table 1). For most of these groups (and under our exploratory analysis), there is no strong statistical reason for the selection of the baseline category (e.g., males versus females). However, with regard to employment, we selected the most commonly selected employment option as the baseline group. For income and education, it was of interest to examine the behaviour of the extreme groups with regard to an "average" socio-demographic group. As some country-level covariates are on a different order of magnitude from each other (e.g., GDP and HDI), all level-2 covariates are scaled to the unit interval using $s(z) = \frac{z - \min(z)}{\max(z) - \min(z)}$ to aid prior specification and model convergence.

Vaccination intent is modelled as $Y_{ij} \sim OrderedLogit\left(\mathbf{x}_{ij}^T \boldsymbol{\beta}_j, (\tau_1, \tau_2, \tau_3)\right)$, where $Y_{ij} \in \{1, 2, 3, 4\}$ is vaccination intent for an individual $i$ in country/territory $j = 1, \dots, J$; $\mathbf{x}_{ij} \in \mathbb{R}^{P \times 1}$ is an (indicator) matrix of $P$ individual-level covariates (which are provided by a binary—or "one hot"—vector according to the sociodemographic status of the individual); $\boldsymbol{\beta}_j \in \mathbb{R}^{P \times 1}$ are individual-level parameters;

and $-\infty < \tau_1 < \tau_2 < \tau_3 < \infty$. We use the ordered logistic distribution for $k$ ordered outcomes specified by $Y \sim OrderedLogistic(\mu; \tau_1, \tau_2, \dots, \tau_{k-1})$, where $P_Y(Y \le k) = \sigma(\tau_k - \mu)$ and $\sigma(x) = 1/(1 + e^{-x})$ is the standard logistic sigmoid function. This definition is equivalent to the proportional-odds assumption, wherein the difference in the log of cumulative odds ratios between successive categories is independent of the slope $\beta$. Individual-level covariates are modelled as $\beta_{jp} \sim t\left(\boldsymbol{\gamma}_p \mathbf{z}_j, \sigma_p^2, 1\right)$ for $p = 1, \dots, P$, where country-level covariates are specified in the matrix $\mathbf{z}_j \in \mathbb{R}^{Q \times 1}$ ($Q$ is the number of country-level covariates); and $\boldsymbol{\gamma} \in \mathbb{R}^{P \times Q}$ is the matrix of fixed-effect parameters. A $t$-distribution is used to allow robust regression of country-specific covariates. Semi-informative normal prior distributions are used for all fixed-effect parameters $\gamma_{pq} \sim N(0, 10)$. Half-normal hyperpriors are placed over variance parameters, $\sigma_p \sim N_+(0, 10)$. These prior widths (specified as variances) place the vast majority of prior mass over plausible parameter values. All individual- and country-level covariates and their recoding are provided in Table 2. To establish the most parsimonious model for vaccine-uptake intent, three subsets of the hierarchical model specified above are fit: (1) an intercept-and-slopes-as-outcomes model (all model parameters, the "full" model); (2) intercepts-as-outcomes (model 1, with $\gamma_{pq} = 0$ if $p > 1$ and $q > 1$); and (3) the "null" model (no level-1 or level-2 covariates)[31]. The model with the lowest deviance-information criterion[32] is determined to be the most parsimonious model.

A Bayesian linear correlation is used to assess the relationship between vaccination intent in the WIN World Survey and 10,822 individuals surveyed from 15 countries in June 2020 from Lazarus[14] (Brazil, Canada, China, Ecuador, France, Germany, India, Italy, Mexico, Nigeria, Poland, South Korea, Spain, United Kingdom, and United States)[14,33]. The survey question in that study differs from the analysis here; respondents were asked to reply to (on a scale from strongly agree to strongly disagree), "you would accept a [COVID-19] vaccine if it were recommended by your employer and was approved safe and effective by the government." The aim of this analysis is to show consistency in estimates for national-level intent to vaccinate, despite variation in survey wording. To show trends in the UK's acceptance of a COVID-19 vaccine, uptake estimates from previous surveys (see Data) are presented with their corresponding credible intervals.

Temporal trends in intent to accept a COVID-19 vaccine are assessed in the United Kingdom before and after the first person was vaccinated with the Pfizer–BioNTech vaccine in the United Kingdom[34] using similar survey data conducted in September[24] ($n = 1000$) and October 2020[11] ($n = 16820$).

All inference is performed via Gibbs sampling, with models implemented in JAGS (using R version 4.0.3). In total, 2000 burn-in iterations and 20,000 iterations were sufficient for parameter convergence for all models. Individual-level parameters are reported as odds ratios (OR) or log odds ratios. All parameters are reported with the corresponding 95% highest posterior-density intervals (95% HPDIs). The 95% HPDI is the smallest interval of the posterior distribution that contains 95% of the probability mass. A breakdown of responses to vaccination intent across all individual covariates for all countries can be found in the supplementary data 1 and 2. Individual-level data and their variable recodings are shown in Table 2. No ethical approval for Lazarus[14] data or WIN's World Survey data was sought as these datasets are in the public domain. Ethical approval for the UK study data was obtained by the Imperial College ethics committee on 15 June 2020 with reference 22130. In the UK dataset, informed consent was obtained from all respondents before they participated in the survey.

**Table 1 Data summary.**

| | | count | Definitely will get vaccinated | | Probably will get vaccinated | | Probably will not get vaccinated | | Definitely will not get vaccinated | | Do not know /No response | |
|---|---|---|---|---|---|---|---|---|---|---|---|---|
| | | | *n* | % | *n* | % | *n* | % | *n* | % | *n* | % |
| Sex | Female | 13673 | 4404 | 32.2 | 4973 | 36.4 | 2464 | 18.0 | 1783 | 13.0 | 49 | 0.4 |
| | Male | 13090 | 4952 | 37.8 | 4720 | 36.1 | 1949 | 14.9 | 1425 | 10.9 | 43 | 0.3 |
| Age | 18–24 | 3593 | 1322 | 36.8 | 1302 | 36.2 | 554 | 15.4 | 403 | 11.2 | 13 | 0.3 |
| | 25–34 | 5623 | 1906 | 33.9 | 2090 | 37.2 | 841 | 14.9 | 756 | 13.4 | 29 | 0.5 |
| | 35–44 | 5034 | 1580 | 31.4 | 1821 | 36.2 | 907 | 1.08 | 700 | 13.9 | 25 | 0.5 |
| | 45–54 | 5041 | 1654 | 32.8 | 1831 | 36.3 | 934 | 18.5 | 607 | 12.1 | 14 | 0.3 |
| | 55–64 | 4131 | 1470 | 35.6 | 1477 | 35.8 | 721 | 17.4 | 459 | 11.1 | 4 | 0.1 |
| | 65+ | 3337 | 1422 | 42.6 | 1170 | 35.1 | 454 | 13.6 | 283 | 8.5 | 7 | 0.2 |
| Employment status | Employed | 16445 | 5786 | 35.2 | 5952 | 36.2 | 2706 | 16.4 | 1943 | 11.8 | 58 | 0.3 |
| | Unemployed | 2955 | 856 | 29.0 | 1076 | 36.4 | 541 | 18.3 | 481 | 16.3 | 0 | 0.0 |
| | Housewife | 2464 | 776 | 31.5 | 903 | 36.7 | 405 | 16.4 | 357 | 14.5 | 23 | 0.9 |
| | Retired/disabled | 2838 | 1233 | 43.4 | 972 | 34.2 | 402 | 14.2 | 225 | 8.0 | 6 | 0.2 |
| | Student | 1826 | 655 | 35.8 | 713 | 39.0 | 307 | 16.8 | 147 | 8.0 | 5 | 0.3 |
| | Refused or do not know (empl.) | 234 | 50 | 21.4 | 77 | 32.9 | 52 | 22.2 | 55 | 23.5 | 0 | 0.0 |
| Highest education | Secondary | 11489 | 3871 | 33.7 | 4209 | 36.6 | 1986 | 17.3 | 1379 | 12.0 | 43 | 0.4 |
| | Primary | 1990 | 662 | 33.3 | 692 | 34.8 | 289 | 14.5 | 332 | 16.7 | 14 | 0.7 |
| | Higher | 12536 | 4603 | 36.7 | 4593 | 36.6 | 2037 | 16.2 | 1272 | 10.2 | 31 | 0.2 |
| | None/other | 553 | 171 | 31.0 | 132 | 23.8 | 60 | 10.8 | 187 | 33.7 | 4 | 0.7 |
| | Refused or do not know (educ.) | 195 | 48 | 24.5 | 67 | 34.5 | 42 | 21.3 | 39 | 19.7 | 0 | 0.0 |
| Income | Medium/low | 15343 | 5234 | 34.1 | 5629 | 36.7 | 2530 | 16.5 | 1895 | 12.3 | 56 | 0.4 |
| | High | 8722 | 3391 | 38.9 | 3158 | 36.2 | 1306 | 15.0 | 844 | 9.7 | 24 | 0.3 |
| | Refused or do not know (inc.) | 2688 | 730 | 27.1 | 902 | 33.5 | 574 | 21.4 | 469 | 17.5 | 13 | 0.5 |
| Health | Unhealthy | 5400 | 1890 | 35.0 | 1963 | 36.4 | 953 | 17.6 | 570 | 10.6 | 24 | 0.4 |
| | healthy | 21136 | 7415 | 35.1 | 7656 | 36.2 | 3417 | 16.2 | 2579 | 12.2 | 69 | 0.3 |
| | do not know (health) | 227 | 51 | 22.3 | 75 | 33.0 | 43 | 18.9 | 59 | 25.9 | 0 | 0.0 |
| Stress | No | 3481 | 1424 | 40.9 | 985 | 28.3 | 455 | 13.1 | 588 | 16.9 | 28 | 0.8 |
| | Yes | 23076 | 7873 | 34.1 | 8651 | 37.5 | 3902 | 16.9 | 2588 | 11.2 | 63 | 0.3 |
| | Do not know (stress) | 206 | 59 | 28.7 | 57 | 27.7 | 56 | 27.2 | 33 | 15.8 | 1 | 0.6 |
| Gov't handling | Badly | 12002 | 3326 | 27.7 | 3950 | 32.9 | 2541 | 21.2 | 2166 | 18.0 | 19 | 0.2 |
| | Well | 13918 | 5906 | 42.4 | 5393 | 38.8 | 1670 | 12.0 | 885 | 6.4 | 63 | 0.4 |
| | Do not know (gov't handling) | 843 | 123 | 14.6 | 350 | 41.5 | 203 | 24.0 | 158 | 18.7 | 10 | 1.2 |

A summary of study factors and raw and breakdown of (weighted) vaccination intent by each covariate used in the study. Note that the total number of weighted responses (26,763) exceeds the total number of respondents (26,759) by four.

**Reporting summary**. Further information on research design is available in the Nature Research Reporting Summary linked to this article.

## Results

**National-level estimates of vaccination intent**. The overall intent to accept a COVID-19 vaccine—that is, those replying that they would either "definitely" or "probably" accept a COVID-19 vaccine—is highest in Vietnam (96.8%, 95% highest posterior-density interval—HPDI—95.3–98.2), India (90.7%, 88.1–93.2), China (90.6%, 88.8–92.3), and Denmark (87.0% 84.1–90.0), and the lowest in Lebanon (44.1%, 39.5–48.2), France (44.0%, 41.1–47.3), Croatia (41.5%, 37.5–45.5), and Serbia (37.8%, 33.6–42.1) (Fig. 1A).

Lebanon (42.6%, 38.4–47.0), Pakistan (31.8%, 28.7–35.1), Paraguay (28.0%, 23.9–32.0), and Serbia (27.8%, 23.9–32.0) have the highest proportion of respondents who state they will "definitely not" take a COVID-19 vaccine (Fig. 1B). Vietnam (69.3%, 65.2–73.0), India (57.8%, 53.5–62.1), Brazil (54.1%, 51.2–57.0), and Mexico (52.9%, 48.4–57.4) have the highest proportion of respondents who say they will "definitely" take a vaccine (Fig. 1B). There are six countries that have a higher proportion of respondents who state they would "definitely not" take the vaccine than would "definitely" take it: Croatia (22.4 versus 13.0%), France (23.9 versus 13.2%), Lebanon (42.8 versus 30.0%), Paraguay (27.9 versus 21.2%), and Poland (21.0 versus 19.6%) (Fig. 1C). Only a slightly higher proportion of

respondents in Pakistan (33.7 versus 31.9%) and Slovenia (20.0 versus 16.3%) would definitely get the vaccine than definitely not (Fig. 1C).

**Associations with vaccination intent**. The intercepts-as-outcomes model (see Methods) had the lowest DIC value among all models. The fixed-effect individual- and country-level determinants of intent to accept a COVID-19 vaccine are shown in Fig. 2A. Only fixed-effect parameters for which the 95% HPDI excludes zero are commented on. Averaged across all countries, males are more likely to accept a COVID-19 vaccine than females (odds ratio—OR—1.34 [95% HPDI 1.21–1.48]), over 65s are more likely to accept than 18–24-year-olds (1.52 [1.26–1.85], while 35-44-year-olds are less likely than 18–24-year-olds (0.84 [0.75–0.95]). (The odds ratio in this context is the odds of moving to a higher level on the ordinal vaccination intent response for the group in question compared with the same odds for the reference group.) Those with higher education (undergraduate or post-graduate degrees) are more likely to state intent to accept than those with secondary education (1.24 [1.14–1.35]). Those unemployed are less likely to state acceptance than those employed (0.88 [0.79–0.98]), but students are more likely (1.19 [1.04–1.36]). High income is associated with increased intent compared to low or middle income (1.23 [1.15–1.33]). A belief that the government is handling the pandemic well is associated with an increased intent to accept the vaccine (2.23 [1.91–2.63]).

**Table 2 Study data for Bayesian ordinal multilevel regressions to establish the determinants of intent to accept COVID-19 vaccines.**

| | Survey item | Values (recode in parenthesis) | Regression baseline |
|---|---|---|---|
| Response variable | **COVID-19 vaccination intent (response)** | | |
| | When a vaccine for the coronavirus becomes available, will you get vaccinated? | Definitely will get vaccinated (4), probably will get vaccinated (3), probably will not get vaccinated (2), definitely will not get vaccinated (1); do not know/no response* (2) | Not applicable (variable is the response) |
| Individual-level covariates | **Socio-demographic** | | |
| | Sex | Male and female | Female |
| | Age | 18–24, 25–34, 35–44, 45–54, 55–64, 65+ | 18–24 |
| | Highest educational attainment | None or only basic education (none/other), completed primary school (primary), completed secondary school (secondary), completed high level of education (higher), completed higher level of education, e.g., master/PhD (higher), other educational level (none/other), refused/do not know/no response (do not know or refused, education) | Secondary |
| | Work status | Working part-time (employed), retired/disabled, student, working full-time (employed), housewife, unemployed, refused/ do not know/ no response (do not know or refused, work status) | Employed |
| | Income | Low (low/middle), medium low (low/middle), medium (low/middle), medium high (high), high (high), refused/ do not know/no response (do not know or refused, income) | Low/middle |
| | Self-reported health | | |
| | How do you consider your overall health in general? | Very healthy (healthy), healthy (healthy), somewhat healthy (healthy), unhealthy, refused/do not know/no response (do not know or refused, health) | Healthy |
| | How often would you say that you suffer from stress? | Very often (often), fairly often (often), sometimes (often), occasionally (not often), never (not often), refused/do not know/ no response (do not know or refused, stress) | Not often |
| | Government handling | | |
| | | Very badly (badly), rather badly (badly), pretty well (well), very well (well), do not know | Badly |
| | **National statistics** | | |
| Country-level covariates | COVID-19 mortality (total deaths per 100,000 population preceding fieldwork) | Continuous variable scaled to the range [0, 1] | n/a |
| | COVID-19 mortality (total deaths per 100,000 population in two weeks preceding fieldwork) | Continuous variable scaled to the range [0, 1] | n/a |
| | Human development index (HDI) 2019 | Continuous variable scaled to the range [0, 1] | n/a |
| | Gross domestic product per capita (GDP) | Continuous variable scaled to the range [0, 1] | n/a |
| | Vaccine confidence | Continuous variable scaled to the range [0, 1] | n/a |

Questionnaire items from WIN World Survey are shown with possible responses and their recodes (individual-level covariates). Country-level covariate data definitions are also shown (country-level covariates). The baseline for the hierarchical ordinal regression (see "Methods") is shown for all covariates. (* A sensitivity analysis is performed to assess the impact that recoding "no response / do not know" to "probably will not get vaccinated" has on our findings, see "sensitivity analysis" and supplementary Figs. 1 and 2 in the supplementary information for further details).

Respondents who did not provide a response or who did not know their employment status or if the government was handling the pandemic well had lower intent to vaccinate than the respective baseline groups of employed and who thought the government was handling the pandemic badly. Countries with a higher more recent death count had a lower overall population acceptance (0.31, 0.13–0.72).

There is some notable variation in country-specific parameters (random effects) about the country-averaged (fixed-effect) parameters. In Fig. 2B, random-effect parameters for which the 95% HPDI excludes zero are shown, thus revealing countries with strong evidence to suggest a link between covariates and vaccine intent. There is evidence to suggest that males are more likely to accept the vaccine in 22 countries (Fig. 2B; SEX), with females in China more likely to report intent to accept the vaccine than males (−0.27, −0.53 to −0.03), and no evidence that males are more likely than females (or vice versa) in nine countries. Over

65 s have a higher vaccination intent in 18 countries, with the strongest effect size in the United Kingdom (1.28, 0.73–1.80) (Fig. 2B; AGE). Paraguay is the only country for which those with primary education are less likely to accept the vaccine than those with secondary education. In 21 countries, higher education (undergraduate or postgraduate degree) is associated with higher vaccination intent than secondary education, with the strongest effect found in the United States (0.53, 0.24–0.80) (Fig. 2B; EDUCATION). Being retired or disabled is associated with increased intent to vaccinate in Croatia, Denmark, Italy, and United States relative to those employed, whereas being unemployed in Chile, Japan, and Peru is associated with decreased intent (Fig. 2B; EMPLOYMENT STATUS). Students in two countries (Ecuador and Ireland) are more likely to accept the vaccine. High income is associated with an increased intent in 24 countries (Fig. 2B; INCOME). Interestingly, the most consistently informative variable of uptake intent is whether the

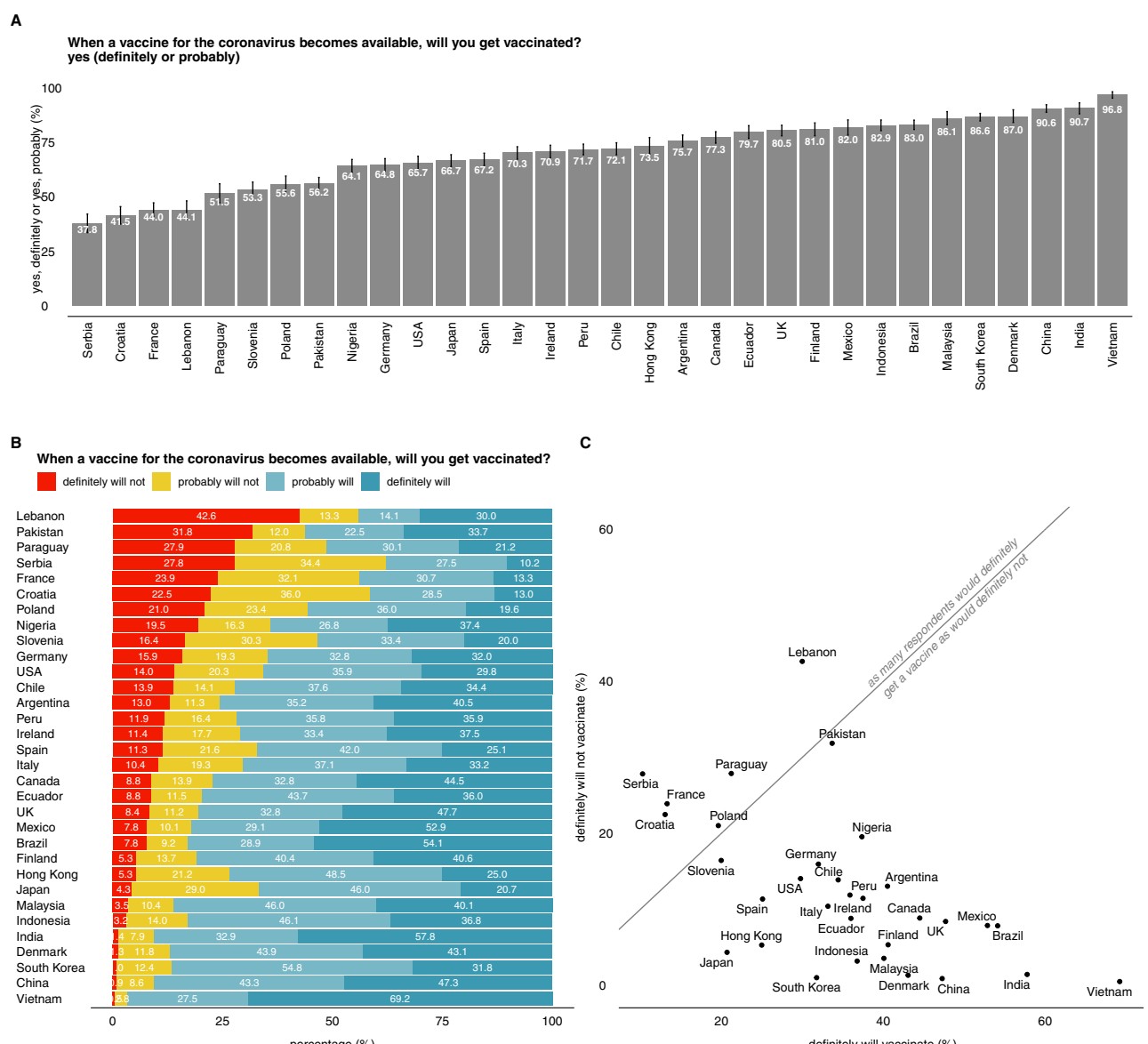

**Fig. 1 Global trends in intent to accept a COVID-19 vaccine. A** Ranking of countries by intent to accept a COVID-19 vaccine, with both positive responses ("definitely will get vaccinated" and "unsure, but probably will get vaccinated") grouped. Error bars denote posterior 95% highest posterior-density intervals. **B** Ranking of countries by the percentage of respondents reporting that they will "definitely" not accept a COVID-19 vaccine, with all survey-response possibilities shown. **C** For each country, the percentage of respondents who would "definitely" vaccinate is shown against the percentage who would "definitely not" vaccinate, thus revealing countries that are polarised in their acceptance of a COVID-19 vaccine. (The diagonal line is y = x and thus shows the line on which as many respondents would definitely accept the vaccine as would not.) Please see supplementary data 1 for all raw counts to vaccination intent.

government is handling the pandemic well. In 27 countries, a belief that the government is handling the pandemic well is associated with higher intent, while in Brazil and the United States a belief that the government is handling the pandemic well is associated with a lower intent, and in Pakistan, reported uncertainty over the government's handling is associated with substantially lower vaccination intent (Fig. 2B; GOV'T HAND-LING). There is no evidence to suggest that self-reported health and self-reported stress were found to be associated with uptake intent. All random-effect parameters are provided in supplementary data 1 and 2.

There is a correlation of $\rho = 0.63$ (95% HPDI, 0.34–0.88) between intent to accept a COVID-19 vaccine in the WIN World Survey data data and those from Lazarus[14], which uses a different question to probe respondents intended uptake if the vaccine was

deemed "safe and effective" by the government (see Fig. 3A and Methods). India has a lower proportion of respondents stating they would take a COVID-19 vaccine in Lazarus[14] (74.5%, 72.4 to 76.6; confidence interval generated assuming outcomes are binomially distributed) than in the WIN data presented here (90.7%, 88.1–93.2). Due to slight differences in survey wording and survey methodology, it is unclear whether this is a survey artifact or a genuine increase in intent to accept the COVID-19 vaccine. Across both studies, India, Mexico, South Korea, and China have the highest stated willingness to accept a COVID-19 vaccine, while France, Poland, and Nigeria have the lowest (Fig. 3A).

There is no evidence to suggest that vaccination views in the United Kingdom have shifted after the introduction of the Pfizer–BioNTech vaccine in the United Kingdom (Fig. 3B); in fact, attitudes stayed remarkably constant between ~4 October

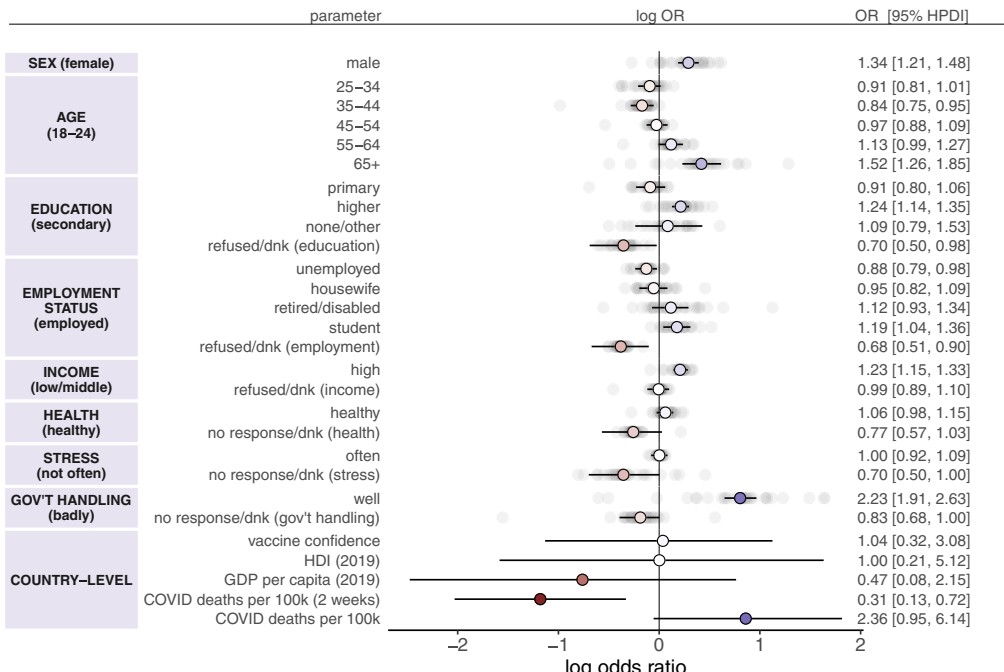

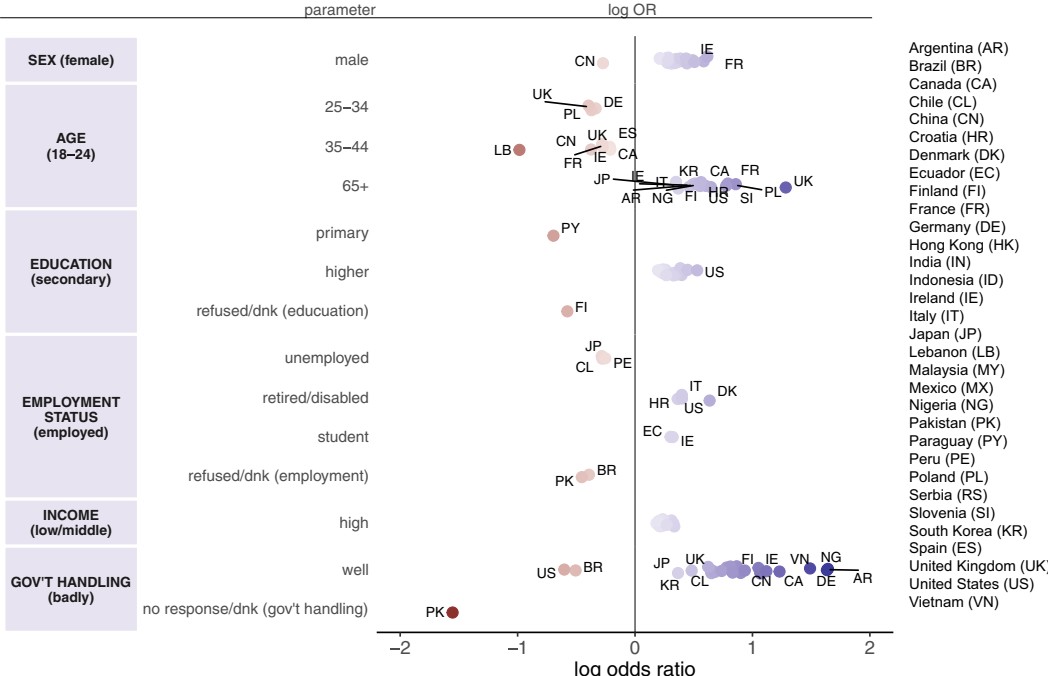

**Fig. 2 Determinants of intent to accept a COVID-19 vaccine. A** The fixed-effect parameters (coloured: reds denote a negative log odds ratio, while blues denote a positive log odds ratio) with 95% highest posterior-density intervals (HPDI; black horizontal bars) from the intercepts-as-outcomes model (see Methods), which represents an "average" of the effects across all 32 countries (greyed). For each parameter, odds ratios and 95% HPDIs are shown (right panel). The left panel denotes the covariate group with the baseline group shown in parentheses. **B** Random-effect parameters (coloured: reds denote a negative log odds ratio, while blues denote a positive log odds ratio) from the intercepts-as-outcomes model. Only parameters whose 95% HPDIs exclude zero are shown (these points are therefore a subset of the greyed points in **A**). (The left panel again shows the covariate group with the baseline in parentheses. Note that not all covariates from (**A**) are shown as some do not contain random-effect parameters for which the 95% HPDI excludes zero.) Countries are labelled with a two-letter abbreviation. Due to crowding of labels, not all data points have labels: the reader should consult supplementary data 1 and 2 for all model parameters.

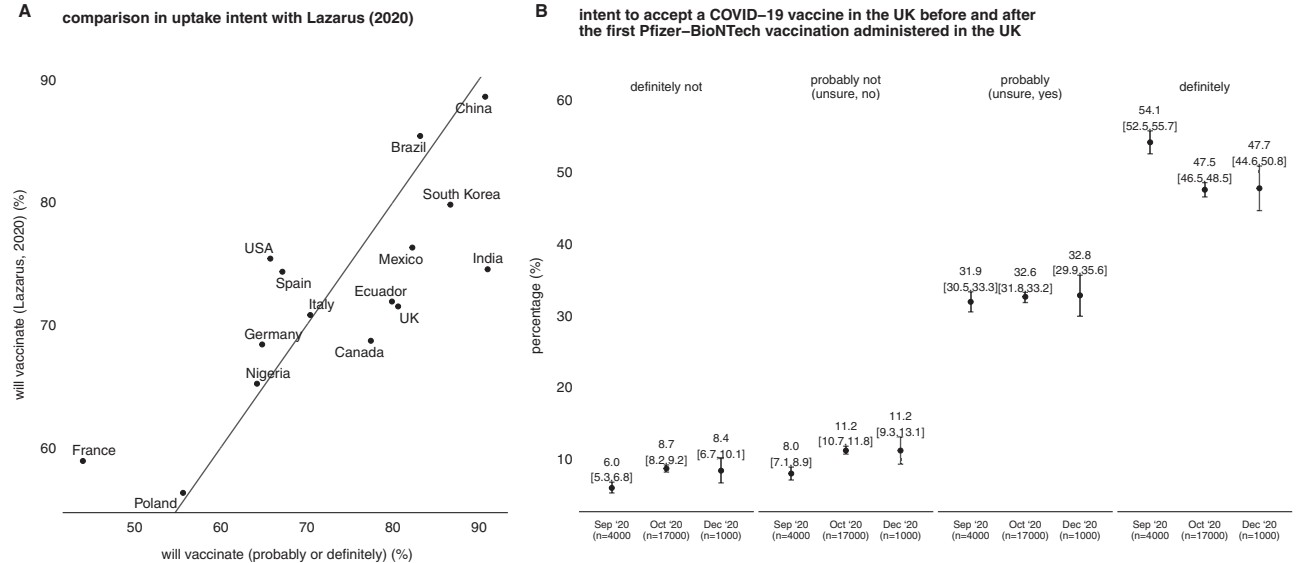

**Fig. 3 Comparison of intent to accept a COVID-19 vaccine in this study to previous global and national surveys.** A A comparison between overall intent to accept a COVID-19 vaccine in this study ("probably or definitely") versus agreement to "you would accept a [COVID-19] vaccine if it were recommended by your employer and was approved safe and effective by the government"[14] (see Methods for further details). **B** A comparison between intent to accept a COVID-19 vaccine in the United Kingdom before (Lazarus[14]; de Figueiredo[3]) and after (this study) both emergency use authorisation by the UK Medicines and Healthcare Products Regulatory Agency on 2 December 2020 and the first vaccination on 8 December 2020 (error bars denote 95% highest posterior-density intervals).

2020 and 11–13 December 2020. In October, 47.5% (95% HPDI, 46.5–48.5) of UK respondents said they would "definitely" take a COVID-19 vaccine compared with 47.7% (44.6–50.8) after vaccine rollout began on 8 December; 32.6% (31.8–33.2) said they were "unsure, but leaning towards yes" in October 2020 compared to 32.8% (29.9–35.8) who said they would "probably" take the vaccine in December; 11.2% (10.7–11.8) said they were "unsure, but leaning towards no" in October 2020 compared with 11.2% (9.3–13.1) ("probably not"); and 8.7 (8.2–9.2) and 8.4 (6.7–10.1) who would not take the vaccine. There was a slight fall, however, in respondents who would "definitely" take the vaccine between September and October 2020 (Fig. 3B).

## Discussion

We conducted a global study on intent to accept a COVID-19 vaccine in 32 countries and examined the individual- and country-level determinants of vaccine uptake intent. There are a number of noteworthy findings. There is substantial cross-country variability in vaccine acceptance, with estimates ranging from 69.3% (65.2–73.0) of respondents in Vietnam who would "definitely" accept the vaccine to only 10.3% (7.6–13.0) in Serbia, 13.0% (10.3–15.8) in Croatia, and 13.0% (11.2–15.5) in France, which consistently has among the lowest vaccine confidence globally[4].

As high as 42.6% (38.4–47.0) of respondents in Lebanon and 31.8% (28.7–35.1) in Pakistan state they would "definitely not" accept a COVID-19 vaccine. Confidence in vaccines has been recorded as falling in the last few years in Pakistan, where a history of vaccine scepticism has posed challenges for polio eradication[20]. Lebanon is only one of five countries surveyed where a belief that the government is handling the pandemic well is not found to be associated with an increased likelihood of vaccinating. A round-the-clock curfew implemented in Lebanon on 14 January has led to protests against the government over rising rates of unemployment[35]. As trust in government is a key driver of vaccine confidence in many settings[36–38], unsupported policies could dent trust in the COVID-19 vaccination programme: this relationship should be monitored closely, especially

in settings with relatively low intent to accept a COVID-19 vaccine or historically low confidence in governments or vaccines.

A belief that the pandemic is being handled well by the government is associated with a lower intent to vaccinate in Brazil and the United States (where the survey was conducted 11–18 November, after the 2020 US general election, but before the inauguration of Joe Biden). In both countries, leaders have previously downplayed the threat of the pandemic (which could, for instance, have affected supporters' views over the importance of a vaccine) and there have been previous questions over whether Trump and Bolsonaro have been vaccine sceptics[39]. While the former leader has seemed to embrace COVID-19 vaccines (although is found to drive vaccine hesitancy[40]), the latter recently stated that "…we're not responsible for any [vaccine] side-effects. If you turn into a crocodile, it's your problem" and has implied that he would not take a COVID-19 vaccine[41]. A recent study in the United States found that those who favour the Republican Party have lower COVID-19 vaccination intent than those who favour the Democrat Party[42], but we can find no studies investigating the link between political preference and COVID-19 vaccine intent in Brazil.

Intent to accept a COVID-19 vaccine appears to be polarised in Poland and Pakistan. In these two countries, as many respondents state they would "definitely" accept the vaccine (21.0%, 17.7–24.4 and 31.8, 28.7–35.1, respectively) than would "definitely not" accept it (19.6%, 16.5–23.0 and 33.7, 31.1–36.1, respectively). Vaccination confidence appears to be declining in both countries[4], resulting possibly from recent and historic misinformation[20,43].

Additionally, we observed sex-, age-, education-, and income-related associations with intent to accept a COVID-19 vaccine. When an association was found between these variables and vaccine intent, it was predominately in the same direction for all countries; however, there were some exceptions, such as females being more likely to report a willingness to accept the vaccine in China (whereas males were more likely to state acceptance in 22 countries). This finding contrasts with a global survey conducted in June 2020[14] that found women to be generally more accepting, when averaged over 19 countries, 14 of which overlapped with this

study. The overall levels of acceptance in these 14 shared countries, however, were found to strongly correlate Fig. 3A. Over 65s, those with high education (undergraduate or postgraduate degrees) and high-income groups were found to have a higher acceptance than 18–24-year-olds and low-/middle-income groups in the majority of countries studied, in alignment with older age groups being more confident in the safety and importance of vaccines in multiple cross-country studies[1,3]. No association was found between self-reported health and stress and vaccination uptake. A recent global review of intended COVID-19 vaccine acceptance measured via multinational and national-level studies identified four countries (Australia, United Kingdom, Ireland, and the United States) where a link between self-reported health and COVID-19 vaccine intent was investigated, but only found a significant link between health condition and vaccine intent in Ireland[13].

We explored the link between national-level uptake intent and national-level factors: we found no evidence of an association between uptake intent and the Human Development Index, GDP per capita, national-level vaccine confidence, and total COVID deaths per 100,000 population. We did, however, find that COVID deaths per 100,000 population in the prior two weeks to fieldwork were negatively associated with uptake intent, that is, that countries with more recent deaths had a lower overall vaccine acceptance. This result is challenging to explain, as we demonstrate (Fig. 3A) that estimates of uptake intent have remained relatively stable for at least 14 of the countries we investigate between June 2020 and October–December, 2020. Countries with high recent COVID-19 deaths at the time of fieldwork included France, Slovenia, and Poland (that have low overall vaccine confidence[4]); while, at the time of fieldwork, China and Vietnam had the lowest two-week cumulative death toll (www.worldometers.info, accessed of 31 January 2021), suggesting that this effect could be dominated by the timing of epidemic peaks in these respective countries.

A comparative study of surveys of vaccine intent in the United Kingdom before and after the local approval of the Pfizer–BioNTech vaccine reveals that the approval and introduction of the vaccine had no impact on vaccination-uptake intent, however, recent uptake figures in the UK of 88.0% for the first dose[44] are notably higher than stated intent among United Kingdom adults in October 2020[11].

There are a number of study limitations to note. Only individual-level covariates available in WIN World Survey were available for analysis, and there may be a large number of factors that could play stronger roles in determining uptake intent if investigated (e.g., direct measurements of COVID-19 vaccine hesitancy or distrust[45], ethnicity[46], risk perception[47], or perceptions of feeling pressured[48]). Further, the estimates of uptake provided here are static and could change drastically in response to factors such as vaccine misinformation. Moreover, the online nature of the surveys in the majority of countries may introduce computer literacy or access biases that may impact the overall estimates of vaccination intent. In countries where computer access and literacy are high, we would expect these biases to be small, but this is nonetheless a source of potential bias for which we have not controlled. Social desirability biases may also impact our estimates of intent to accept COVID-19 vaccines that may also vary by mode of questionnaire (e.g., online versus telephone surveys)[49]. We have not attempted to control for these potential biases but we do note that October 2020 survey of intent to vaccinate in the United Kingdom made predictions about vaccine uptake that slightly underestimated observed uptake[50].

A robust communication system that engages with the public over issues of distrust and safety can help support acceptance of COVID-19 vaccination and contribute to confidence building around existing routine immunisation programmes, many of which have been disrupted in the context of the COVID-19 pandemic. Tailored efforts to listen to and address the concerns and issues among underserved or marginalised groups, will be crucial to ensuring equity in national vaccination efforts. Losses of trust in one vaccine can quickly drive uptake losses in other— unrelated—immunisations[4]. Robust monitoring systems for vaccine confidence and misinformation can provide for the early detection of losses in system confidence. Confidence issues can then be addressed via clear communication strategies tailored to specific socioeconomic or cultural groups that will respond differently to both the message and type of messenger.

### Data availability

All raw WIN World Survey data are available at https://winmr.com/win-world-survey-covid19-vaccine-and-intention-to-travel-in-2021/, while cleaned data used in this study can be found at https://osf.io/8vezs/. All data from Lazarus[14] are available at https://osf.io/kzq69. The UK data are from a previous large-scale study[11]. All source data for the article figures can be found in supplementary data 3.

### Code availability

The custom code to implement multilevel ordinal regressions can be found at https://osf.io/8vezs/[30].

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

## Acknowledgements

AdF and HJL would like to thank ORB (Gallup) International for providing access to the WIN World Survey data. This study is unfunded.

## Author contributions

AdF designed and implemented the statistical analyses, performed analyses, created figures, and wrote the paper. AdF and HJL interpreted the results.

## Competing interests

AdF and HJL are involved in Vaccine Confidence Project collaborative grants with GlaxoSmithKline and Janssen Pharmaceutica outside the submitted work. HJL has also received honoraria as a member of the Merck Vaccine Confidence Advisory Board and GlaxoSmithKline advisory roundtables. HJL is a member of the Merck Vaccine Confidence Advisory Board.
