## [Peer Review File · Communications Medicine]

Reviewers' comments:

Reviewer #1 (Remarks to the Author):

Dear authors,

Thank you for your study investigating determinants of COVID-19 vaccine acceptance in 32 countries and comparing attitudes over time in a subset. It contributes to a small and growing set of studies on this topic and is timely for policymakers, researchers, and other readers. The article is well written with a strong methodology.

Minor Comments:

- Discussion, p9. The first sentence of the concluding paragraph seems to have a structural issue.

Reviewer #2 (Remarks to the Author):

This is a valued and timely set of findings derived from a rich and valuable data resource. The number of countries included is impressive and the graphical representation of results makes the findings very accessible to the reader. While I believe the study overall is an important one, the coding of 'don't know' responses as 'probably won't get vaccinated' is a significant concern that potentially undermines the analyses and findings. I hope the authors find my comments useful.

Abstract

1. Information relating to the data used, and the methodology/analytic strategy employed would be helpful and informative from the outset.

Introduction

2. In the second sentence the authors write "addressing the demand side of COVID-19 vaccines needs equal attention". The authors might exercise caution here. This paper specifically investigates intention to vaccinate. 'Vaccine demand' is a distinct phenomenon and is distinguishable from intention and vaccine hesitancy/acceptance. Moreover, the authors seem to use 'intent', 'acceptance' and 'demand' interchangeably throughout the manuscript. I would advise that they aim for specificity and consistency throughout to avoid unnecessary confusion.

3. I assume 'income ethnicity' is a typo. Also, a thorough proof read of the entire manuscript is required.

4. The authors have not stipulated testable hypotheses. Given that many of the indicators included have previously and consistently been associated with vaccine intent/acceptance/hesitancy, predictions would seem possible/sensible.

Results

5. The first section provides methods content and should be relocated.

6. Can the authors cite evidence for "and because there is a high collinearity between national-level confidence in the importance of vaccines for children and in other national-level confidence measures in the safety and effectiveness of vaccines"?

7. "Temporal trends in intent to accept a COVID-19 vaccine are assessed in the UK before and after the first person was vaccinated with the Pfizer-BioNTech vaccine in the UK27 using similar survey data conducted in September22 (n=1000) and October 202023 (n=16820)." This is repetitive – see

end of intro.

8. Under 'National-level estimates of vaccination intent' – can the authors clarify that they have chosen to report the top 4 countries in each category. Also, can the authors provide 95% Confidence Intervals for these estimates?

Discussion

9. The authors write “there may be a large number of factors that could play stronger roles in determining uptake intent if investigated”. Can the authors direct their readers to the rich literature that exists in relation to these other factors?

10. Also, the authors might wish to consider limitations relating to mode and reliability of data collection i.e. “mode effects” – see Zhang et al., (2017). “...many studies examine data quality and the effects of social desirability when using different survey methods. In some studies, computer surveys yielded similar results as paper and pencil surveys, e.g., on attitude questionnaires (Booth-Kewley, Edwards, & Rosenfeld, 1992) or for personally sensitive questions (Knapp & Kirk, 2003). In other studies, however, different results were found when using different survey methods, e.g., on satisfaction-dissatisfaction questions (Dillman et al., 2008) or on questions about consumption frequency and preferences related to wine (Szolnoki & Hoffmann, 2013). Furthermore, response biases for telephone interviews and internet questionnaires caused by social desirability have been reported (Chang & Krosnick, 2009). Here, more social desirability was manifested for telephone compared to Internet surveys, respectively. Some studies also showed that biases related to social desirability tended to be highest for telephone surveys and lowest for web surveys (Holbrook et al., 2003, Kreuter et al., 2008).”

Method

11. A brief synopsis of the WIN World Survey would be very helpful for those who may be unfamiliar with this rich data resource.

12. Can the authors identify the countries that used telephone interviews (four, n=2803), telephone-assisted web interview (one, n=600), and face-to-face interviews (two, n=2500)?

13. The authors write that “Bayesian hierarchical ordinal logistic regression is used to probe the ties between individual- and country-level covariates”. Can the authors explicate what variables were individual and what variables were country-level please? Can the authors also describe the coding of these covariates please?

14. The authors write that “A total of 92 respondents responded that they “do not know” whether they would get a COVID-19 vaccine. To avoid the loss of missing data, these 90 responses are recoded to “probably won’t get vaccinated” as they demonstrate hesitancy about vaccinating, but no strong intent to reject the vaccine.” First, is the difference between 92 and 90 a typographical error? If not, please clarify. Second, only four possible responses were detailed in the data section. Was ‘don’t know’ a fifth option? If so, please amend data section content. Third, given that response options included ‘probably will’ and ‘probably not’ why have the authors elected to recode these responses in the negative? The rationale provided is unconvincing.

15. The authors write that “The baseline group for individual-level covariates is....”. Do the authors mean that the reference categories for the individual covariates were....? Can the authors explain why these reference categories were selected? Also, how are nominal income categories ‘low, medium, high’ to be interpreted as quintiles?

References

Zhang, X., Kuchinke, L., Woud, M. L., Velten, J., & Margraf, J. (2017). Survey method matters: Online/offline questionnaires and face-to-face or telephone interviews differ. *Computers in Human Behavior*, 71, 172-180.

Reviewer #3 (Remarks to the Author):

The authors report a Bayesian analysis of a WIN survey of 26,759 individuals across 32 countries between October 21 and December 15, 2020.

The authors explore both individual- as well as country-level covariates using an elegant Bayesian approach which should be commended. The paper is a welcome departure from the classical frequentist approaches used to analyze survey data.

I only have a couple of issues that should be commented upon:

1. It would have been interesting to see whether the results would have changed had the authors combined answers 3 and 4 into one response "negative towards vaccination". I would recommend including this as a sensitivity analysis.
2. How did the authors choose the mean of the normal distribution? Why square root of 10 and not, say, pi? I am not saying the approach is incorrect but it should be substantiated by previous research,
3. Why does the t-distribution of beta μ have only 2 degrees of freedom?
4. Please add the number of responders and the duration of the study in the abstract.
5. Several non-standard abbreviations are used in the text, such as WIN, HDI, HPDI, without being spelled out first. Please adjust.

Reviewer #4 (Remarks to the Author):

The manuscript entitled 'Global intent to accept COVID-19 vaccinations' is a report of the intention to accept a COVID-19 vaccine in 32 countries and the authors examine the individual- and country-level determinants of the intention to get vaccinated.

The strength of the manuscript is the impressively high number of countries involved in the data set. Thus, the manuscript is a valuable descriptive global report of a single-item measure. However, the rather random selection of predictors of the intention to get vaccinated, the missing theoretical foundation of the models, the lack of relevant predictors and control variables from previous research about vaccine intent and vaccine hesitancy and the explorative rather than confirmative nature of the study are major weaknesses of the manuscript. I will explore these points in more detail below.

- 1) The authors use age, education, gender, income, employment status, perceived health, perceived stress and perceived government performance as predictors for the individuals' intention to get vaccinated. The selection of predictors seems to follow no rational and the introduction does not provide any hint of why these predictors were selected or why they are deemed relevant. Behavioural intentions, such as vaccination intentions, are researched in various social sciences and the general theoretical models (for example: Theory of Planned Behaviour, Health Belief Model) and vaccine hesitancy specific models (for example: 3C Model by the SAGE working group) provide

important scientific guidance on relevant predictors of vaccination intention. The theoretical models on vaccination intention could have been used to inform the introduction, to generate items for the questionnaire or at least to describe potential limitations of the study. However, no health behavior theory was used to inform the study nor was any of the theories mentioned. In fact, there is little use of measuring only sociodemographic variables in the research field of vaccination intentions and hesitancy. The explanatory value of these variables is very limited and conclusions that might be used to design interventions or policies or recommendation are thus equally limited.

2) The selection of perceived stress and perceived health status are simply coming out of nowhere. Why is perceived stress measured in the first place and is this single-item solution of measuring stress a valid and reliable indicator of an individual's stress level?

3) The limited explanatory value of the predictors used in this study is also reflected in the discussion. The discussion lacks specific conclusions for researchers or practitioners that can be derived from the presented data. The conclusion starts with 'A robust communication system that engages with the public over issues of distrust potential safety fears can not only help support acceptance of COVID-19 vaccination but can contribute to confidence building...'. I agree with all of the sentences provided in the conclusion but I do not see how this concluding paragraph is informed by the data presented. What does the reader learn specifically from your data? What is the added valued, potential recommendation that can be directly derived from the results section?

4) The ORs reported in the Result section should be interpreted in context. The authors do not provide any sort of classification of the size of the effects in relation to conventions or findings in other studies. It could also help to simply provide additional sentences on what an OR of XX means for a younger adult compared to an older adult etc.

5) The main primary outcome was measured with only four points. Thus, individuals needed to position themselves whether they are rather pro or anti. In fact, the selection of a scale with an even number of points may make the countries and individuals look more polarized than they actually are. This concern is underlined by the fact that individuals who did state that they do not know (n =92) were included in the anti-group without a clear rationale. Thus, intentions are somewhat treated as a binary phenomenon which may not reflect the global sentiment.

6) Were the statistical models and/or the method preregistered? Were there any aprior hypotheses or is this a mere exploratory study? These limitations should be mentioned if they apply.

7) The limitation section should also discuss that the majority of responses was received from online samples.

8) The authors state 'The survey question in that study differs slightly from the analysis here; respondents were asked to reply to (on a scale from strongly agree to strongly disagree), "you would accept a [COVID-19] vaccine if it were recommended by your employer and was approved safe and effective by the government."' This question appears very different to me because the question includes the premise that the vaccine is recommended by an employer and approved safe. This was not included in the question reported in Table 2 and should make a difference because of the importance of official recommendations when analysing behavioural intentions.

9) Please clarify what the regression baseline of n/a (response) means for the primary outcome in

Table 2. The selection of a baseline for predictors (e.g. female for gender) is intuitive. However, I do not understand the rational for the response variable.

Reviewer 1	
Dear authors, Thank you for your study investigating determinants of COVID-19 vaccine acceptance in 32 countries and comparing attitudes over time in a subset. It contributes to a small and growing set of studies on this topic and is timely for policymakers, researchers, and other readers. The article is well written with a strong methodology.	We thank the reviewer for their kind comments.
Minor Comments: - Discussion, p9. The first sentence of the concluding paragraph seems to have a structural issue.	We have fixed the structural issue in the first sentence of the final paragraph of the discussion as recommended.

Reviewer 2	
This is a valued and timely set of findings derived from a rich and valuable data resource. The number of countries included is impressive and the graphical representation of results makes the findings very accessible to the reader. While I believe the study overall is an important one, the coding of 'don't know' responses as 'probably won't get vaccinated' is a significant concern that potentially undermines the analyses and findings. I hope the authors find my comments useful.	We thank the reviewer for their comments that will undoubtedly help strengthen our manuscript. We agree that the coding of 'do not know' responses to 'probably will not get vaccinated' appears to be arbitrary. We believe that there are benefits to this re-classification; namely, that respondents who state that they 'do not know' whether they would accept a COVID-19 vaccine would likely fall between the two extremes 'definitely will' and 'definitely will not' get vaccinated. The choice of 'probably will not get vaccinated' therefore errs on the side of higher hesitancy (so we do not overstate intent to vaccinate in a given country). However, to allay any fears that this reclassification (among a relatively small fraction of the sample) impacts results, we have now completed two sensitivity analysis: (1) coding 'no response' values into 'probably will get vaccinated'; (2) a complete-case analysis that removed 'no response' values from the analysis. In total there are only 92 'do not know / no response' values (this represents far less than 1% of the total sample size). The sensitivity analyses reveal no substantive change in the quantitative or qualitative interpretation of the fixed-effect parameters. We have now included a small supplementary appendix which details our sensitivity analysis and its conclusions. In short, there is one parameter. With regards to the interpretation of 'significant results' there is one fixed-effect parameter which we deem as 'significant' in the main text, which become 'non-significant' under the sensitivity analysis. (Though we point out that (a) these changes in HPDI are extremely small and (b) the selection of a 95% HPDI to obtain 'significant' variables is non-standard in a Bayesian setting). The parameter for 'no response / do not know' for whether individuals believe their Government has handled the pandemic well (GOV'T HANDLING) moves from 0.83 (0.68 to 1.00) in the main text to 0.84 (0.69 to 1.02) in the complete case sensitivity and 0.83 (0.68 to 1.01) in the recoding to 'probably will get vaccinated'. Further, in the main text, the fixed-effect parameter for 55-64-year-olds is 1.13 (0.99 to 1.27), whereas in the complete case analysis this becomes 1.13 (1.00 to 1.27). We signpost this result in the main text.
Abstract 1. Information relating to the data used, and the methodology/analytic strategy employed would be helpful and informative from the outset.	We note that we are already on the abstract limit and therefore we probably would not get away with providing too much detail here, although we acknowledge that the reviewer is right that some early insight to the methods would be instructive to the reader. We have now made a slight tweak to the abstract to help in this regard.
Introduction 2. In the second sentence the authors write "addressing the demand side of COVID-19 vaccines needs equal attention". The authors might exercise caution here. This paper specifically investigates intention to vaccinate. 'Vaccine demand' is a distinct phenomenon and is distinguishable from intention and vaccine hesitancy/acceptance. Moreover, the	We thank the reviewer for their comments, especially those on consistent (and accurate) terminology. We have removed all reference to 'demand' and instead focus on vaccine acceptance. Income ethnicity was indeed a typo, and we have fixed this, along with a few other typos/grammatical bugs throughout.

authors seem to use 'intent', 'acceptance' and 'demand' interchangeably throughout the manuscript. I would advise that they aim for specificity and consistency throughout to avoid unnecessary confusion. 3. I assume 'income ethnicity' is a typo. Also, a thorough proof read of the entire manuscript is required.	
4. The authors have not stipulated testable hypotheses. Given that many of the indicators included have previously and consistently been associated with vaccine intent/acceptance/hesitancy, predictions would seem possible/sensible.	We did not stipulate a testable hypothesis. Indeed, in answer to many of the feedback points raised by reviewer 4, we recommend viewing this study as exploratory and—to this end—we have stated this wherever relevant in the manuscript.
Results 5. The first section provides methods content and should be relocated.	We thank the reviewers for this and have moved this chunk to the discussion. (Although we find it a bit odd to dive straight into the results without a description of the survey questions at a bare minimum: so we have left these in, and perhaps the editors can advise on what would be the best way of presenting this portion of the manuscript?)
6. Can the authors cite evidence for “and because there is a high collinearity between national-level confidence in the importance of vaccines for children and in other national-level confidence measures in the safety and effectiveness of vaccines”?	Done!
7. “Temporal trends in intent to accept a COVID-19 vaccine are assessed in the UK before and after the first person was vaccinated with the Pfizer-BioNTech vaccine in the UK27 using similar survey data conducted in September22 (n=1000) and October 202023 (n=16820).” This is repetitive – see end of intro.	Thank you for this spot, we have lessened this repetition
8. Under 'National-level estimates of vaccination intent' – can the authors clarify that they have chosen to report the top 4 countries in each category. Also, can the authors provide 95% Confidence Intervals for these estimates?	We're a tad confused by this feedback point. We do specify 95% HPDIs. If the reviewer is referring to the fact we have not specified what HPDI is, or given a description, we thank the reviewer for spotting that this was missing and this has now been rectified.
Discussion 9. The authors write “there may be a large number of factors that could play stronger roles in determining uptake intent if investigated”. Can the authors direct their readers to the rich literature that exists in relation to these other factors?	Yes, we really should have included references at this point. We have now rectified this.
10. Also, the authors might wish to consider limitations relating to mode and reliability of data collection i.e. “mode effects” – see Zhang et al., (2017). “...many studies examine data quality and the effects of social desirability when using different survey methods. In some studies, computer surveys yielded similar results as paper and pencil surveys, e.g., on attitude questionnaires (Booth-Kewley, Edwards, & Rosenfeld, 1992) or for personally sensitive questions (Knapp & Kirk, 2003). In other studies, however, different results were found when using different survey methods, e.g., on satisfaction-dissatisfaction questions (Dillman et al., 2008) or on questions about consumption frequency and preferences related to wine (Szolnoki & Hoffmann, 2013). Furthermore, response biases for telephone interviews and internet questionnaires caused by social desirability have been reported (Chang & Krosnick, 2009). Here, more social desirability was manifested for telephone compared to Internet surveys, respectively. Some studies also showed that biases related to social desirability tended to be highest for telephone surveys and lowest for web surveys (Holbrook et al., 2003, Kreuter et al., 2008).”	This is something we have in fact considered for a more recent study we have prepared. In fact, with regards to the UK, a previous manuscript that forecasted vaccination uptake in the UK actually slightly underpredicted uptake. This might suggest that social desirability biases are small (at least in the UK context). However, the level of discussion we developed was quite sophisticated around this point, and we feel that may be a bit out of place here. Instead, we add a citation to that study (currently a pre-print), but also use your suggested citation (which is very useful) to express possible limitations around social desirability. We thank the reviewer for raising this important limitation.
Method 11. A brief synopsis of the WIN World Survey would be very helpful for those who may be unfamiliar with this rich data resource.	Yes, we also agree this would be useful, and have now added a brief description with citation in the Methods section
12. Can the authors identify the countries that used telephone interviews (four, n=2803), telephone-assisted web interview (one, n=600), and face-to-face interviews (two, n=2500)?	This information is provided in the extended table, which sets out the country, the survey methodology, sample size, national coverage, and fieldwork dates. However, we omitted this reference from the main text, which has now been included! We thank the reviewer for spotting this.

13. The authors write that “Bayesian hierarchical ordinal logistic regression is used to probe the ties between individual- and country-level covariates”. Can the authors explicate what variables were individual and what variables were country-level please? Can the authors also describe the coding of these covariates please?	Table 2 shows all variables with their recoding as well as setting out which variables are individual-level and which are country-level (see left-most column in Table 2). However, to make sure the reader is reminded, we have now included an extra link to this table following the paragraph which contains the text to which the reviewer refers.
14. The authors write that “A total of 92 respondents responded that they “do not know” whether they would get a COVID-19 vaccine. To avoid the loss of missing data, these 90 responses are recoded to “probably won’t get vaccinated” as they demonstrate hesitancy about vaccinating, but no strong intent to reject the vaccine.” First, is the difference between 92 and 90 a typographical error? If not, please clarify. Second, only four possible responses were detailed in the data section. Was ‘don’t know’ a fifth option? If so, please amend data section content. Third, given that response options included ‘probably will’ and ‘probably not’ why have the authors elected to recode these responses in the negative? The rationale provided is unconvincing.	We thank the reviewer for picking up on these issues. Firstly, it was actually 95 respondents: 92 (not 90, which was a typo) is the total number of weighted ‘do not know / no response’ respondents, and we had simply pulled this number from table 1 forgetting that this was weighted data (we have now added labels on this table to stress that these data are weighted values). Secondly, we would refer to reviewer to our feedback to reviewer 1 who provides the same concern. We have provided a full sensitivity analysis that both (a) removes all ‘do not know / no response’ responses and (b) recodes ‘do not know / no response’ into ‘probably will get vaccinated’. Our results are effectively the same under this sensitivity, with some extremely minor changes to note.
15. The authors write that “The baseline group for individual-level covariates is....”. Do the authors mean that the reference categories for the individual covariates were....? Can the authors explain why these reference categories were selected? Also, how are nominal income categories ‘low, medium, high’ to be interpreted as quintiles?	There is no overwhelming reason for us to choose specific baseline groups, other than it makes sense to choose groups that are (a) well populated and (b) that facilitate a comparison of interest. For example, the choice between males and females is arbitrary, while we choose employed as this is the most commonly selected option. We have attempted to provide clarification on this in the manuscript. Income is recoded from a quintile to high versus medium/low and we have twice (as far as we can tell) accidentally incorrectly referred to these new codes as ‘quintiles’ which we have now fixed.

Reviewer #3	
The authors report a Bayesian analysis of a WIN survey of 26,759 individuals across 32 countries between October 21 and December 15, 2020. The authors explore both individual- as well as country-level covariates using an elegant Bayesian approach which should be commended. The paper is a welcome departure from the classical frequentist approaches used to analyze survey data. I only have a couple of issues that should be commented upon:	We thank the reviewer for comments that will undoubtedly strengthen our manuscript.
1. It would have been interesting to see whether the results would have changed had the authors combined answers 3 and 4 into one response “negative towards vaccination”. I would recommend including this as a sensitivity analysis.	We are not sure what this would add by way of a sensitivity analysis as, in dichotomising, we are losing information. This could result in parameters becoming ‘non-significant’ but this does not imply that there is not an association between a covariate and response. I suppose another way of seeing this would be to pose the question “if you had continuous data, can you perform a sensitivity analysis by dichotomising the data?” Afterall, ordinal regression is binary regression if the number of ordinal categories is equal to two. (See, e.g, ‘A simulation study evaluating approaches to the analysis of ordinal outcome data in randomized controlled trials in traumatic brain injury: results from the IMPACT project’ Clin Trials 2010.) If the reviewer has a reference of where this has been implemented and the benefits it offers, we will gladly take a look!
2. How did the authors choose the mean of the normal distribution? Why square root of 10 and not, say, pi? I am not saying the approach is incorrect but it should be substantiated by previous research,	The mean of all our prior distributions is zero, which is standard practice. We agree that we have not been sufficiently clear in our methods section regarding the choice of priors, including why $\sqrt{10}$ was chosen as the standard deviation of our normal distributions. To clarify this point, the variance is equal to 10 in these prior distributions: this allows considerable prior mass over plausible parameter values. We have now stated this in our methods.

3. Why does the t-distribution of beta β have only 2 degrees of freedom?	Thank you for this spot. This is a typo and should say '1'. This prior is useful in this modelling context as we do not wish to perform too much shrinkage on parameters that are far from the global mean, as the associations between a socio-demographic factor on uptake may be consistent in n-1 countries but may differ substantially in 1 country. This prior choice better guards against this eventuality.
4. Please add the number of responders and the duration of the study in the abstract.	This change has now been implemented
5. Several non-standard abbreviations are used in the text, such as WIN, HDI, HPDI, without being spelled out first. Please adjust.	We thank the reviewer for these spots. We have now defined all of these abbreviations in the first instance which they are mentioned in the manuscript.

Reviewer #4 The manuscript entitled 'Global intent to accept COVID-19 vaccinations' is a report of the intention to accept a COVID-19 vaccine in 32 countries and the authors examine the individual- and country-level determinants of the intention to get vaccinated. The strength of the manuscript is the impressively high number of countries involved in the data set. Thus, the manuscript is a valuable descriptive global report of a single-item measure. However, the rather random selection of predictors of the intention to get vaccinated, the missing theoretical foundation of the models, the lack of relevant predictors and control variables from previous research about vaccine intent and vaccine hesitancy and the explorative rather than confirmative nature of the study are major weaknesses of the manuscript. I will explore these points in more detail below.	We thank the reviewer for their comments, and we believe they will lead to much more clarity, especially with regards to the nature of our study being exploratory rather than being based on pre-specified hypotheses. To quickly address some of the points above:  ■ We completely agree that our paper lacks theoretical foundation; we had no control over the questionnaire design or data collection. We believe it is best to view this study as an 'exploratory study' and – to this end—we have attempted to clarify that this is so at key early points in the manuscript (such as the abstract and introduction). ■ We would respectfully disagree that the variables we have used are irrelevant. In fact, socio-demographic factors alone have been able to do a good job at forecasting COVID-19 vaccination intent in the UK (de Figueiredo, Forecasting trends in COVID-19 vaccine uptake in the UK, Medrxiv, 2021: see validation study therein). These variables are often the variables that policymakers are foremostly interested in when they contact us (the Vaccine Confidence Project) about possible public health interventions. (We address this specific point in more depth below.)
1) The authors use age, education, gender, income, employment status, perceived health, perceived stress and perceived government performance as predictors for the individuals' intention to get vaccinated. The selection of predictors seems to follow no rational and the introduction does not provide any hint of why these predictors were selected or why they are deemed relevant. Behavioural intentions, such as vaccination intentions, are researched in various social sciences and the general theoretical models (for example: Theory of Planned Behaviour, Health Belief Model) and vaccine hesitancy specific models (for example: 3C Model by the SAGE working group) provide important scientific guidance on relevant predictors of vaccination intention. The theoretical models on vaccination intention could have been used to inform the introduction, to generate items for the questionnaire or at least to describe potential limitations of the study. However, no health behavior theory was used to inform the study nor was any of the theories mentioned. In fact, there is little use of measuring only sociodemographic variables in the research field of vaccination intentions and hesitancy. The explanatory value of these variables is very limited and conclusions that might be used to design interventions or policies or recommendation are thus equally limited. 2) The selection of perceived stress and perceived health status are simply coming out of nowhere. Why is perceived stress measured in the first place and is this single-item solution of measuring stress a valid and reliable indicator of an individual's stress level?	Unfortunately, we had no control over survey design or data collection. Nonetheless, we believed that this dataset represented an excellent opportunity to explore global trends in COVID-19 vaccine intentions. We believe that we are reasonably clear about this point, but we have now stressed this throughout the manuscript. If the reviewer has further suggestions on how we might communicate this point (including on stress), we will happily act on this. With regards to the use of only socio-demographic variables in the field of vaccine intentions and hesitancy, we would have to disagree quite firmly with this point. Taking the UK as an example, since there are significant relationships between a range of socio-demographic predictors and intent to vaccinate [and now vaccine uptake¹] (most strikingly, with respect to age and Black/Black British ethnicities²⁻⁴), then this information can be used to forecast uptake at sub-national levels across the UK. Indeed, AdF has successfully forecasted COVID-19 vaccine uptake across the UK in October 2020 (that is, before a vaccine was even approved in the UK) using only demographic variables (de Figueiredo, Forecasting trends in COVID-19 vaccine uptake in the UK, Medrxiv, 2021). This research has been used by policymakers in the UK (NHS/PHE/DHSC) to plan vaccine rollout and communication drives. So, we believe that socio-demographic variables alone can be exceptionally useful. With regards to whether the socio-demographic drivers are useful in this study, the VCP has been actively engaged with a range of national and international policymakers regarding the socio-demographic determinants of intent to vaccinate. These analyses often provide a 'first-port-of-call' to understand the nature of

	hesitancy issues via, e.g., focus groups to better understand confidence concerns among high hesitancy socio-demographic groups.
3) The limited explanatory value of the predictors used in this study is also reflected in the discussion. The discussion lacks specific conclusions for researchers or practitioners that can be derived from the presented data. The conclusion starts with 'A robust communication system that engages with the public over issues of distrust potential safety fears can not only help support acceptance of COVID-19 vaccination but can contribute to confidence building...'. I agree with all of the sentences provided in the conclusion but I do not see how this concluding paragraph is informed by the data presented. What does the reader learn specifically from your data? What is the added value, potential recommendation that can be directly derived from the results section?	We, in part, would point to our response above. From experience we know that policymakers often contact us about—in the first instance at least—the socio-demographic determinants of interest to them. This forms a guide to them about where hesitancy may arise in their country (e.g. geographically). Policymakers can then, for example, test communication strategies among these low confidence groups. Our data outlines quite clearly what socio-demographic groups may be of particular risk of low uptake in different settings. This is crucially important in an epidemic context: often low confidence [low uptake] individuals can cluster geographically precisely because they share similar socio-demographic traits. This clustering can disproportionately increase required vaccination levels for herd immunity in adjacent regions⁵. In the UK context, knowing that Black / Black British communities are far less willing to vaccinate than Whites has a direct impact on vaccination policy and epidemic risk. Many parts of the country with relatively high fractions of Black/Black British ethnicities may therefore warrant targeted interventions via (e.g.) communication/immunisation drives.
4) The ORs reported in the Result section should be interpreted in context. The authors do not provide any sort of classification of the size of the effects in relation to conventions or findings in other studies. It could also help to simply provide additional sentences on what an OR of XX means for a younger adult compared to an older adult etc.	It is extremely challenging to compare the value of ORs between different studies when the control variables study (from a purely robust statistical point of view). We have compared many of our findings to directions of associations in other studies, and we have now further strengthened this with more citations with regards to the association between age and vaccine confidence. We have also added a definition of the odds ratio in the context of ordinal regression in the main text.
5) The main primary outcome was measured with only four points. Thus, individuals needed to position themselves whether they are rather pro or anti. In fact, the selection of a scale with an even number of points may make the countries and individuals look more polarized than they actually are. This concern is underlined by the fact that individuals who did state that they do not know (n =92) were included in the anti-group without a clear rationale. Thus, intentions are somewhat treated as a binary phenomenon which may not reflect the global sentiment.	We are a tad confused by the claim that a scale with an even number of points would exaggerate polarisation. The ordinal scale allows respondents to assign themselves on vaccination intent, rather on some continuous scale, which we think is preferred in the context of elucidating intentions. And our statistical methodology is specifically not-binary. We consider polarisation to be a function of when there are equal numbers in the extreme vaccine intention camps. Of course, polarisation in this context lacks a robust definition and theoretical background, and we do not seek to introduce either here. We are merely using the data to establish countries with an equal number of 'definitely's and 'definitely not's' This will naturally establish problematic countries. We feel that polarisation is a fair term in this context, but we would happily rephrase if the reviewer thinks an alternative would be more appropriate.
6) Were the statistical models and/or the method preregistered? Were there any aprior hypotheses or is this a mere exploratory study? These limitations should be mentioned if they apply.	The study was not registered and there were no hypotheses. This study was exploratory and we hoped to have clarified this in our revised manuscript.
7) The limitation section should also discuss that the majority of responses was received from online samples.	We agree this could be a source of notable bias and we have included a limitation note on this in the discussion as the reviewer suggests.
8) The authors state 'The survey question in that study differs slightly from the analysis here; respondents were asked to reply to (on a scale from strongly agree to strongly disagree), "you would accept a [COVID-19] vaccine if it were recommended by your employer and was approved safe and effective by the government.'" This question appears very different to me because the question includes the premise that the vaccine is recommended by an employer and approved safe. This was not included in the question reported in Table 2 and should make a difference because of the importance of official recommendations when analysing behavioural intentions.	We thank the reviewer for bringing this to our attention. Our aim was of course not to conceal this to our attention. Our aim was of course not to conceal this to our attention! We have tried to be explicit when comparing these different phrasings. As table 1 is reserved for the parameters in our main analysis (that is, the multilevel regressions), we do not want to include this question as it may cause some reader confusion regarding that analysis. Instead (and we hope this is to the satisfaction of the reviewer), we (a) have amended the caption of table 1 to reflect the above and (b) have added additional clarifying text about this comparison in the results and methods. Our intention in Figure 3A – rather than to make any claims about official recommendations – is to highlight that intent to accept

	a COVID-19 vaccine (as measured in our survey) is in agreement with intent to accept a vaccine from another survey, even with different wordings – thus showing consistency with the existing literature.
9) Please clarify what the regression baseline of n/a (response) means for the primary outcome in Table 2. The selection of a baseline for predictors (e.g. female for gender) is intuitive. However, I do not understand the rationale for the response variable.	'n/a' here simply means 'not applicable' (since the variable is the response not a predictor, and therefore does not have a reference group. We have now clarified this in the table, and thank the reviewer for bringing this to our attention.

REVIEWERS' COMMENTS:

Reviewer #2 (Remarks to the Author):

All points have been satisfactorily attended to. I wish the authors well with their ongoing research.

Reviewer #3 (Remarks to the Author):

No further comments, accept.

Reviewer #4 (Remarks to the Author):

I thank the authors for their responses.

Most of my initial points have been addressed. The manuscript has improved by adding additional clarifying information such as the label "exploratory" where applicable.

The authors do not agree with my general critique about the decision of using socio-demographic variables only. I agree that these variables can be strong predictors but they provide little additional insights into our scientific understanding of vaccine hesitancy. Even if ethnicity may predict hesitancy it is not the reason why individuals refuse vaccination. However, this is a more general concern and the authors do not claim to provide theory driven work and make clear that the selection of questions was limited by design and may still be helpful for health providers.

Thus, I have no further comments and congratulate to this piece.